# Image Segmentation and Quantification of Droplet dPCR Based on Thermal Bubble Printing Technology

**DOI:** 10.3390/s22197222

**Published:** 2022-09-23

**Authors:** Mingjie Zhu, Zilong Shan, Wei Ning, Xuanye Wu

**Affiliations:** 1School of Microelectronics, Shanghai University, Shanghai 200444, China; 2Shanghai Industrial µTechnology Research Institute, Shanghai 201800, China

**Keywords:** droplet dPCR, droplet localization recognition, image vignetting correction, signal clustering

## Abstract

Thermal inkjet printing can generate more than 300,000 droplets of picoliter scale within one second stably, and the image analysis workflow is used to quantify the positive and negative values of the droplets. In this paper, the SimpleBlobDetector detection algorithm is used to identify and localize droplets with a volume of 24 pL in bright field images and suppress bright spots and scratches when performing droplet location identification. The polynomial surface fitting of the pixel grayscale value of the fluorescence channel image can effectively compensate and correct the image vignetting caused by the optical path, and the compensated fluorescence image can accurately classify positive and negative droplets by the k-means clustering algorithm. 20 µL of the sample solution in the result reading chip can produce more than 100,000 effective droplets. The effective droplet identification correct rate of 20 images of random statistical samples can reach more than 99% and the classification accuracy of positive and negative droplets can reach more than 98% on average. This paper overcomes the problem of effectively classifying positive and negative droplets caused by the poor image quality of photographed picolitre ddPCR droplets caused by optical hardware limitations.

## 1. Introduction

Digital Polymerase Chain Reaction (dPCR) is the promising third-generation nucleic acid detection technology that is playing an increasingly important role in disease diagnosis, individualized medicine, and genetic testing. dPCR has a lower detection limit than RT-qPCR and is more tolerant to inhibitors, which can be used in the detection of SARS-CoV-2 at low concentrations with reduced incidence of false-negative results [1,2,3]. dPCR technology can be used for accurate detection of fetal aneuploidy [4,5], and fetal-free DNA in maternal plasma can be analyzed to detect genetic diseases such as thalassemia and hemophilia [6,7]. dPCR technology can detect tumor cells and DNA fragments transferred to the patient’s blood and body fluids, enabling dynamic monitoring of the patient’s disease course [8,9,10]. dPCR performs dilution of sample DNA molecules to disperse them into different micro reaction units, ultimately with only one or a limited number of DNA molecules in each partition, reducing the competition between target and non-target DNA molecules and improving the sensitivity of the assay [11,12]. Traditional 96/384-well plate analysis cannot meet the requirements of dPCR for the number of reaction partitions, and the microliter scale of reagents for reaction units results in high reagent consumption costs [13]. Micro-processing technology provides an efficient method to generate a large number of microchambers for dPCR with microfluidic chips. Droplet Digital PCR (ddPCR) uses droplets as reaction units, which is an ideal dPCR platform as it is easier to miniaturize the equipment than the integrated microfluidic chip [14,15,16]. The detection sensitivity of droplet digital PCR technology often depends on the total number of reaction units, and the higher the number of reaction units, the better the sensitivity and accuracy of the analysis [17]. In this paper, 24 pL micro reaction droplets are generated with a thermal bubble inkjet printing chip, which achieves miniaturization and high throughput of the reaction system. The droplets after PCR reaction are imaged by an optical system, and the positive and negative droplets are classified and counted by combining with an image analysis workflow to avoid statistical difficulties arising from manual counting.

Flow cytometry and imaging are two commonly used methods in ddPCR to quantify and classify droplets [18,19]. The droplet classification and quantification method used in this paper is the imaging method, in which a CCD camera in the optical platform takes images of droplets in the result reading chip for analysis, and each position of the result reading chip is imaged under the bright-field and fluorescence channels, respectively, and a total of 100 sets of images are taken. The bright-field images are used for droplet identification and localization. Then, the localization information is used in the fluorescence images for droplet segmentation and signal statistics. Finally, positive and negative droplets are classified according to the fluorescence signal intensity [20,21]. However, due to the fabrication variation or handling issue, the images captured by the optical module contend spots and scratches in the background, or even partially defocused images, which affects the analysis of droplet signals [22]. The images captured in the fluorescence channels are not only affected by the inherent optical parameters of the CCD camera, and the image vignetting, but also by autofluorescence particles [23,24]. To achieve accurate identification and localization of droplets in the bright field image, the image analysis workflow in this paper suppresses bright spots and scratches in the bright field image, compensating for the vignetting of the fluorescence channel image, calculates the signal value of each droplet in the image. In the end, positive and negative droplets are classified and counted the droplets using the k-means clustering algorithm [25]. The classification and quantification of high-throughput droplets are achieved in the case of poor flow channel plate process and limited imaging quality of the optical module.

## 2. Materials and Methods

### 2.1. High-Throughput Droplet Generation and Acquisition of Image Data

To generate 24 pL droplets, a new droplet generation method is used. ddPCR droplets are generated by a generator controller controlling a thermal bubble inkjet print chip. The droplet-generating chip is a MEMS thermal bubble actuator realized by a semiconductor process [26,27,28,29]. A small volume of sample solution, normally 20 µL, is pipetted into an inkjet print chip, and the print controller generates hundreds of thousands or even millions of droplets at the 24 pL level in a short time and injects them into the manifold, saving reagent costs and achieving high droplet throughput. The droplets are fed into the optical module for image acquisition after 40 cycles of reaction in the thermal cycler. The entire droplet PCR process is shown in Figure 1a.

The result reading chip of fixed size is used to conduct CCD camera imaging at the fixed position of the optical system and each result reading chip results in 100 images. The motion control module of the optical system can determine the *x* and *y* coordinates of each image at the position of the result reading chip, move each region of interest to the specified coordinate position, and adjust the *z*-axis to the focus distance so that all the regions of the result reading chip containing droplets is scanned. The position information of each image obtained was used for imaging under a bright field and all fluorescence channels, and the image was saved for subsequent image analysis. Figure 1b shows the steps of image acquisition of the optical module. The blue box in Figure 1b shows a large picture composed of 100 bright-field images and one bright-field image (The red box represents one of the 100 images taken). The large picture is composed of 100 fluorescence channel images and one of them is displayed in the green box.

### 2.2. Droplet Identification, Localization, and Noise Filtering

The bright-field image is characterized by the outer edge of the droplets, that identifies and localizes the droplets (Figure 2a). We chose the feature point detection method of SimpleBlobDetector provided in OpenCV to segment and locate droplets [30]. The algorithm first sets a series of continuous thresholds to convert the input gray image into multiple binary images. Equation (1) represents the set of multiple binary images with threshold range [T1, T2] and step size t. Then, the boundary of each binary image is detected to form a connected domain and each boundary center is calculated. All central points are combined and classified into a series of groups (speckle features). Finally, the localization and size information of each group of the gray image is determined.
(1)[T1,T1+t,T1+2t,T1+3t,…T2].

The fabrication of the result reading chip will affect the droplet image quality, reduce the droplet segmentation accuracy and affect the droplet quantification results. Particle debris, bright spots, or scratches on the result reading chip should be suppressed to improve the accuracy of droplet identification and positioning. The bright spots in bright-field images appear to have high gray value and largely connected domain area in the images. In droplet localization and identification, threshold processing, and connected domain area, convexity and roundness screening methods can be used to screen out the connected areas that do not meet the requirements to suppress bright spots and scratches [31,32,33,34]. The detected spot area was limited to [500, 2000] pixels, droplets were screened and bright spots were suppressed by setting the roundness and convexity of spots. The roundness of spots is shown in Equation (2), and the convexity is shown in Equation (3) [35,36]. The value range of the two is 0–1. Area, Perimeter, and convex area indicate the area, perimeter, and convex hull corresponding to each spot. The droplets identified after screening are shown in Figure 2b, and the bright spots in the green circle are suppressed.
(2)$$\mathrm{Circularity}=\frac{4\pi (\mathrm{Area},A)}{{\left(\mathrm{Perimeter},P\right)}^{2}}.$$
(3)$$\mathrm{Convexity}=\frac{\mathrm{Area}}{\mathrm{Convexarea}}.$$

Some background impurities in the green box in Figure 2c are incorrectly identified as droplets. The intensity of the gray value of background impurities in the 16-bit fluorescence image is about 50% of the intensity of the gray value of negative droplets, and the background impurities can be suppressed on the fluorescence image by setting the threshold. The 100 bright-field images and 100 fluorescence images were processed according to the above steps, and we stitched the processed images for easy observation, and the results are shown in Appendix A.

### 2.3. Droplet Segmentation in the Fluorescent Image and Signal Quantification

Inherent defects in the optical system and angular deviations in the light received by the CCD camera can affect the uniformity of dPCR fluorescence images and result in image vignetting (the causes of vignetting are explained in Appendix A) [37,38,39]. To reduce the effect of the camera’s inhomogeneous light sensitivity on the droplet signal values, the vignetting generated by the optical module is compensated to improve the accuracy of dPCR positive and negative droplet classification. The vignetting compensation steps are as follows:Gaussian filter operator is used to blurring an image of the uniform droplet in the fluorescence channel. The blurred image (Figure 3a) is used as the vignetting correction template. The 3 × 3 Gaussian template g(x, y) is shown in Equation (4) and multiplied by a constant 1/16 to achieve a weighted average of the pixel values in the 3 × 3 neighborhood [40].
(4)$$\begin{array}{cc} g(x,\;y)= & \frac{1}{16}\{f(x-1,\;y-1)+f(x-1,\;y+1)+f(x+1,\;y-1)+f(x+1,\;y+1)\;\\ & +[f(x-1,\;y)+f(x,\;y-{1)+f(x+1,\;y)+f(x,\;y+1)]}^{{}^{2}}{+\;f(x,\;y)}^{4}\}. \end{array}$$

2.A quadratic polynomial is chosen as the fitting formula, and the pixel gray value surface is fitted to the template image using the least-squares method, and the surface fitting function is shown in Equation (5) [41,42]. The vignetting correction template is sampled to find the surface fitting coefficients, and the fitted surface is drawn as shown in Figure 3b. The maximum value of the fitted surface equation is divided by the surface fitting value of the image, and the result is normalized to obtain the normalized dark corner surface compensation model as shown in Figure 3c.


(5)
$${c(x,\;y)=a}_{0}\;{+\;a}_{1}{x+a}_{2}{y+a}_{3}x^{2}\;{+\;a}_{4}{\mathrm{xy}+a}_{5}y^{2}.$$


3.The calculated vignetting compensation model is used to correct the images captured by the camera to eliminate image vignetting. Figure 3d shows the image without vignetting correction, and the gray value of the image is counted along the yellow diagonal line in Figure 3d and the gray value curve is plotted as shown in Figure 3e. The image after vignetting correction and the gray value curve of the image are shown in Figure 3f,g. Comparing Figure 3e,g, it is obvious that the image vignetting is better compensated. The pixel value surfaces of the images before and after vignetting correction are shown in Appendix A. The average values of the difference between the average pixel values of the eight edge regions and the average pixel values of the center region in the attached Appendix A are calculated as shown in the attached Appendix A, and the results are 667 and 71, respectively, and the uniformity of the images after vignetting correction significantly improved. The satellite droplets affect the accuracy of the fluorescence field droplet signal calculation, and the pixel values of each droplet region are sorted and selected as the signal value of this droplet by averaging the 40–60% length part of the sequence, and Appendix A represents the segmented droplet signal image after vignetting correction, and the satellite droplets are marked in the orange box.

The vignetting corrected images are automatically classified into two classes of positive and negative droplets using the unsupervised learning k-means clustering algorithm. k-means algorithm achieves the classification of positive and negative droplets mainly by minimizing the value of the intra-cluster variance [43]. The scatter plot of the k-means algorithm clustered positive and negative droplet signals is shown in Figure 4a, and the clustering results of positive droplets and negative droplets are labeled in Figure 4b,c, respectively, in the fluorescence channel images, where the positive droplets clustering results are labeled with red circles in Figure 4b and the negative droplets clustering results are labeled with blue circles in Figure 4c. The quantitative results were obtained by counting the number of positive droplets and negative droplets.

## 3. Results

### 3.1. Bright-Field Image Droplet Segmentation Accuracy

The sample solution is prepared and the droplets are generated for PCR reactions with an inject-based droplet generator, the images are acquired using an optical system, and finally, the droplets are classified and quantified by an image analysis workflow.

1, 5, 10, and 20 brightfield images were randomly selected from the sample and applied three different workflows to count the number of droplets identified and calculate the accuracy of droplet segmentation. The image segmentation algorithm was applied to segment the brightfield images to obtain the number of segmented droplets in each image. After marking the outer contour of droplets on the original image, the droplets with segmentation errors and droplets without effective segmentation are manually observed and counted (Appendix A shows the droplets with segmentation errors and droplets without effective segmentation, the incomplete droplets at the edge of the image are not counted). The actually segmented droplets are the number of droplets correctly segmented in the image minus the incorrectly segmented droplets. The total number of droplets in the image is the sum of the actual number of segmented droplets, the number of droplets incorrectly segmented, and the number of droplets that are not effectively segmented.

In this work, we composed four image analysis workflows. Workflow 1 used fixed threshold screening, kirsch edge detection operator processing, fixed threshold image binarization, morphological processing, and area screening; Workflow 2 changed fixed threshold image binarization to adaptive threshold binarization screening and added ellipse detection methods on the basis of Workflow 1; Workflow 3 added watershed algorithm for secondary segmentation on the basis of Workflow 2. The number of droplets identified in 1, 5, 10, and 20 bright field images was calculated and the accuracy of droplet segmentation was calculated for Workflow 1, 2, and 3, and the results are shown in Table 1. (The specific flow of the three types of workflows is shown in Appendix A, and the segmentation results are shown in the attached Appendix A).

Using the SimpleBlobDetector algorithm to count the number of droplet recognition and calculate the accuracy of droplet segmentation. Compared with the other three workflows, the accuracy of this bright-field droplet image segmentation algorithm is higher, and the actual correct segmentation recognition rate of droplets in the image reaches more than 99% on average, as shown in Figure 5a. combined with Appendix A, it can be seen that when the droplet imaging in the bright field image is blurred, the other three workflows appear in the image in areas where many droplets are not recognized, and the algorithm stability is not high. Meanwhile, we also compared our outcome to different workflows and third party software (Appendix A).

### 3.2. Fluorescence Image Droplet Classification Accuracy

Droplets recognized and localized in bright field images (Figure 5a) were used to remove background in blank regions, scratches, and particles for the fluorescent image segmentation (Figure 5b). The image analysis workflow can calculate the difference between positive droplets (Figure 5d) and negative droplet (Figure 5f) clustering center values to filter out the droplet images with a poor clustering effect.

We further counted the true positives (TP), false negatives (FN), false positives (FP), and true negatives (TN) of the 20 images. We also calculated accuracy, sensitivity, and specificity for the 20 images, and higher values of the three indicate better validity of classification performance, and their calculation formulae are shown in (6)–(8).
(6)$$\mathrm{Accuracy}=\frac{\mathrm{TP}+\mathrm{TN}}{\mathrm{TP}+\mathrm{FN}+\mathrm{FP}+\mathrm{TN}}.$$
(7)$$\mathrm{Sensitivity}=\frac{\mathrm{TP}}{\mathrm{TP}+\mathrm{NF}}.$$
(8)$$\mathrm{Specificity}=\frac{\mathrm{TN}}{\mathrm{FP}+\mathrm{TN}}.$$
where accuracy indicates the ratio of the number of correctly classified droplets to the total number of droplets, sensitivity and specificity indicate the proportion of correctly classified positive droplets to all positive droplets, and the proportion of correctly classified negative droplets to all negative droplets, respectively. The statistical means of accuracy, sensitivity, and specificity for the 20 images were 99.75%, 95.32%, and 99.95%.

### 3.3. Validation of Positive Droplet Concentration Gradient Detection with Multiple Fluorescence Channels

A series of positive droplet concentrations were prepared by mixing positive droplets containing three dyes (Fam, Hex, Cy5) all at a concentration of 50 nM with negative droplets without dye at different ratios. The positive droplet and negative droplet solutions were mixed in a 4:5 ratio and diluted sequentially to produce four samples with a concentration gradient of 44.44%, 27.50%, 17.18%, and 10.72% of positive droplets, and finally, the four samples were analyzed and the actual measured positive droplet ratios were obtained.

The experimental procedure above was repeated four times under the same experimental conditions, and a total of 48 sets of experimental data were obtained for three channels. The mean value of each concentration measured in each fluorescence channel obtained in the four experiments was counted, and the standard deviation of the mean value of the four measured concentration gradients from the actual concentration was calculated. The correlation curves for the three channels are plotted in Figure 6 and the standard deviation for each concentration is labeled, where the actual measured concentration of the algorithm is strongly correlated with the ideal concentration, with correlation coefficients of 0.9937, 0.9764 and 0.9977 for the Fam, Hex and Cy5 channels, respectively.

## 4. Discussion

The droplet inkjet printing chip in this paper can generate more than 100,000 micro-droplets with a volume of 24 pL, and the inkjet printing system can generate high-throughput droplet micro-reaction units within a second, which is conducive to improving the sensitivity of ddPCR detection. The recognition and classification of droplets are accomplished by means of optical module photography and image processing.

In our work, we developed a method to generate droplets with the diameter of only 32.5 µm. Due to the small size of our droplets, we noticed that the effect of the scratches, dirt particles, and vignetting, etc., on droplet segmentation and fluorescent intensity readout are more significant compared to large droplets with the diameter over 100 µm. This paper uses the SimpleBlobDetector algorithm to recognize droplets and removes the bright spot problem in the bright field image by setting the parameters of droplet area and roundness in the recognition process and suppresses the background contamination by setting the image threshold range in the fluorescence channel to improve the accuracy of droplet recognition. We compared the droplet recognition accuracy of the SimpleBlobDetector method used in this paper with that of traditional methods such as threshold detection and watershed segmentation and found that our workflow is more stable and has higher droplet recognition accuracy, and the effective droplet segmentation accuracy of the speckle detection algorithm used is higher than 99% for 20 randomly selected bright field images. We also compared our outcome to different workflows and third party software. Our image analysis workflow showed higher accuracy and more stable performance.

There is vignetting in the fluorescence field images taken by the optical module, and the vignetting can affect the calculation of the intensity of the droplet signal values and reduce the accuracy of the classification. In this paper, a polynomial surface fitting method is used to calculate the vignetting template of the camera and compensate for the captured image to eliminate the vignetting problem of the image. The interference of satellite droplets with the signal value was suppressed by taking the average of the gray value of the middle 20% of each droplet as the signal value of the droplet. The gray value range was set for each fluorescence channel to suppress abnormal bright spots in the fluorescence channel, and then the signal values of each droplet counted were clustered and analyzed to obtain the number of positive and negative droplets. The statistical accuracy of droplet classification on 20 fluorescence images is higher than 98%, which verifies that the image analysis workflow can accomplish the classification and quantification of high-throughput droplets in the case of poor flow channel plate process and limited photographic accuracy of the optical module.

Bright-field images may have vignetting caused by inaccurate focus in the process of capture, which affects the recognition and localization of droplets by the SimpleBlobDetector algorithm. To increase droplet identification and localization efficiency, the algorithm will be improved to compensate for the focus blurry phenomenon in bright field images. In addition, this paper studied Gaussian surface fitting and polynomial fitting when calculating the vignetting compensation of the fluorescence channel, and other fitting methods can be found to improve the accuracy of vignetting correction. We selected K-means clustering for the classification of positive and negative droplets, and other clustering methods can be explored to improve the classification accuracy. In this paper, satellite droplets appear in the droplets printed by the inkjet printing chip. By improving the droplet homogeneity and reducing the influence of satellite droplets on the signal value, we can also improve the accuracy of the initial concentration calculation of the sample.

## Figures and Tables

**Figure 1 sensors-22-07222-f001:**
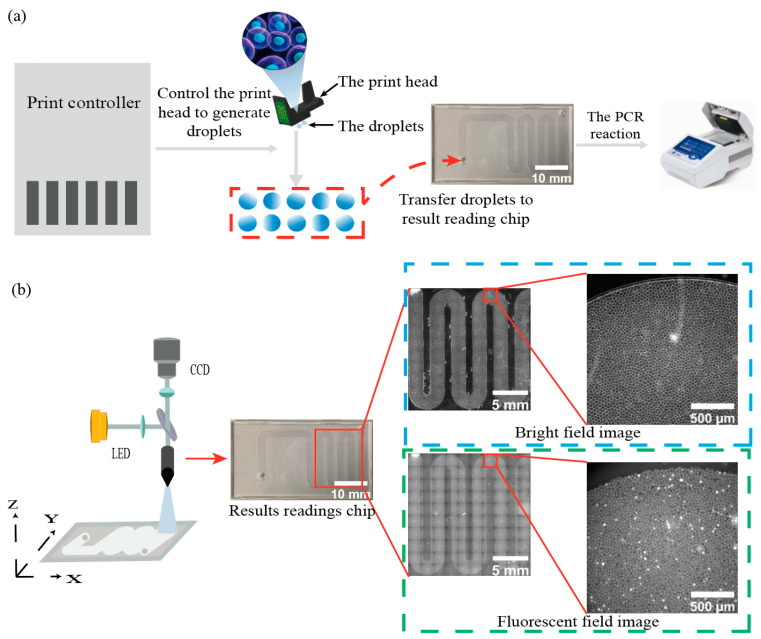
Acquisition of dPCR image data with the high-throughput droplet generation and result reading chip. (**a**) Flow diagram of a thermal bubble inkjet print head generating dPCR droplets, where the print controller controls the droplet generation chip to generate hundreds of thousands of micro-droplets at the 24 pL level, and the generated droplets are transferred to a result reading chip and subjected to PCR reactions in the PCR instrument. (**b**) Flow chart of the optical module taking pictures and imaging the result reading chips after the PCR reaction, fixing the CCD camera, and moving result reading chips in a small area each time, the optical module will image under the bright field and fluorescence field under the corresponding result reading chips position.

**Figure 2 sensors-22-07222-f002:**
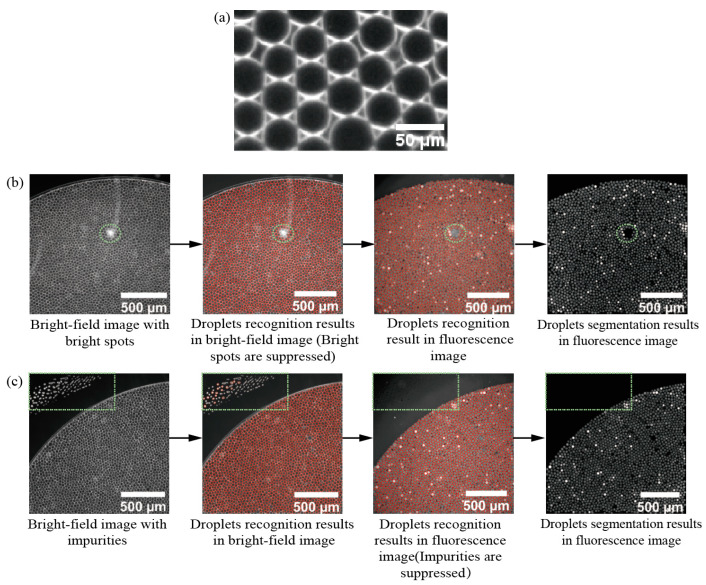
Localization, identification, and suppression of imaging noise in the result reading chip (the dotted circles in the figure represent bright spots or scratches, and the dotted boxes represent background impurities in the result reading chip). (**a**) The droplet outline is clearly shown in the bright-field image. (**b**) Suppression of bright spot contamination in images. The SimpleBlobDetector algorithm is applied to suppress bright spots by setting parameters such as area, roundness, and convexity of droplets in bright field image droplet recognition. (**c**) Suppression of impurity contamination in images. The intensity of the gray value of background impurities in the 16 bit fluorescence image is about 50% of the intensity of the gray value of negative droplets, and the background impurities can be suppressed on the fluorescence image by setting the threshold.

**Figure 3 sensors-22-07222-f003:**
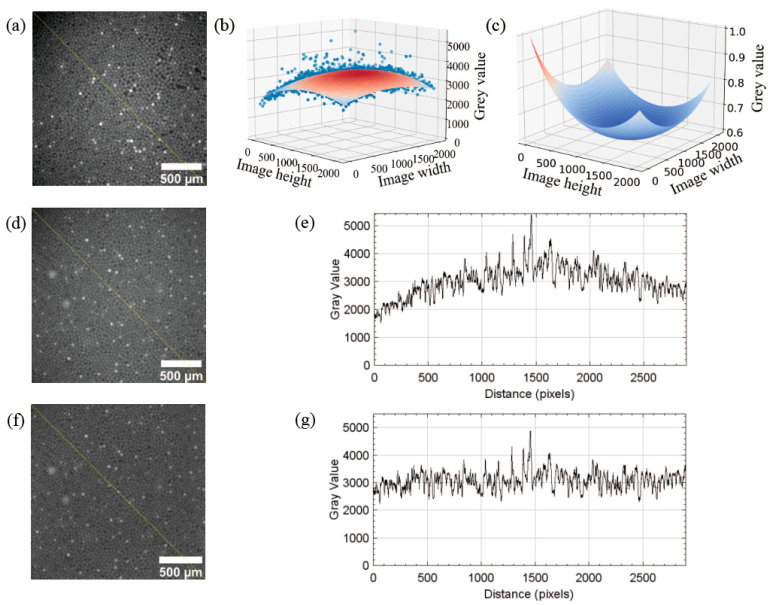
Quantification of droplet signal intensity value in the fluorescence image. (**a**) The template image was used for vignetting correction. (**b**) The fitting surface was obtained from sampling and polynomial fitting of the asymptotic halo compensation image. (**c**) The calculated normalized vignetting compensation model. (**d**) A droplet image of the fluorescent channel without accretion correction. (**e**) The gray value curve is plotted along the yellow diagonal in (**d**). In the absence of asymptotic halo correction, the gray value curve shows a peak shape as a whole. (**f**) The image of (**d**) after vignetting correction. (**g**) The gray value curve is drawn along the yellow diagonal in (**f**). After vignetting correction, the grayscale curve is flat and the image vignetting is corrected.

**Figure 4 sensors-22-07222-f004:**
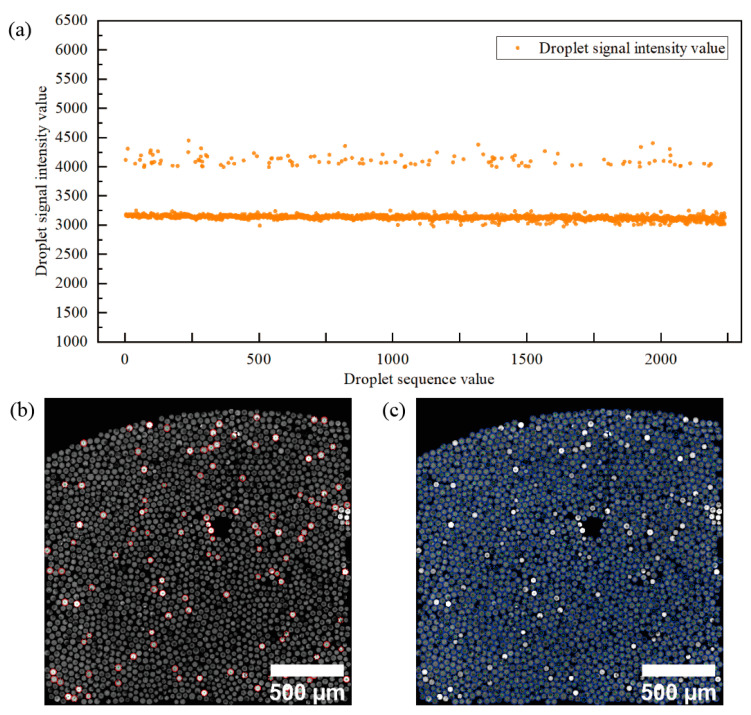
K-means algorithm for classification and quantification of negative and positive droplets. (**a**) the scatter plot of the positive droplet and negative droplet signals after clustering by the k-means algorithm. (**b**) The result of labeling positive droplets on the fluorescence channel image. The number of red circles indicates the number of positive droplets in the image. (**c**) The result of labeling negative droplets on the fluorescence channel image. The number of blue circles indicates the number of negative droplets in the image.

**Figure 5 sensors-22-07222-f005:**
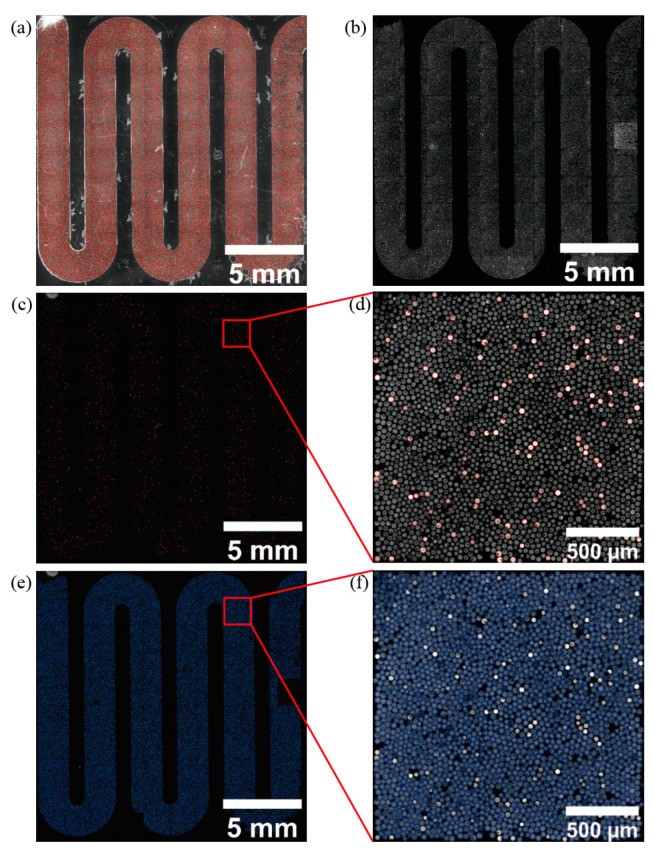
Plots of classification results for positive and negative droplets. (**a**) The result of droplet localization for bright-field images. (**b**) The resulting map of the bright-field image localization information in the fluorescence channel image segmented droplets. (**c**) The resulting map of positive droplets was identified and classified. (**d**) The resulting map of positive droplets (red circles) is identified from a fluorescence image in (**c**). (**e**) The resulting map of negative droplets was identified and classified. (**f**) The result of identifying negative droplets (blue circles) of a fluorescent image in (**e**).

**Figure 6 sensors-22-07222-f006:**
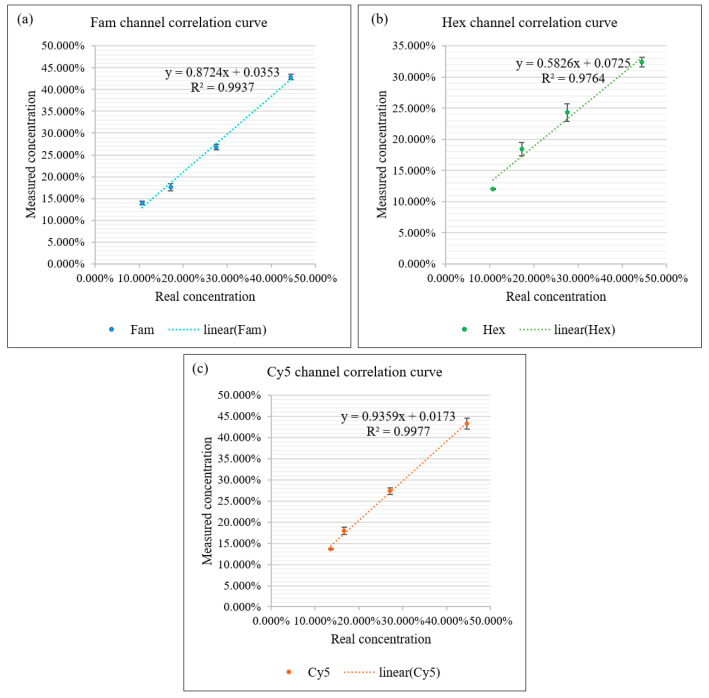
Correlation curves and standard errors of measured and true values of concentration gradients for the three fluorescence channels. (**a**) Correlation curves and standard errors of measured and true values for four concentrations in Fam channel. (**b**) Correlation curves and standard errors of measured and true values for four concentrations in the Hex channel. (**c**) Correlation curves and standard errors of measured and true values for four concentrations in Cy5 channel.

**Table 1 sensors-22-07222-t001:** Accuracy of bright-field image droplet segmentation for the above three segmentation workflows and workflow with SimpleBlobDetector algorithm.

Number of Bright Field Images	Total Number of Droplets	Number of Droplet Segmentation	Number of Droplets Actually Segmented	Number of Segmentation Errors	Segmentation Accuracy (%)
Workflow 1					
1	1917	1859	1843	16	96.14%
5	8092	7724	7612	112	94.07%
10	17,165	16,102	15,797	305	92.03%
20	33,136	29,100	28,223	877	85.17%
Workflow 2					
1	1917	1771	1764	7	92.02%
5	8092	7214	7150	64	88.36%
10	17,165	14,869	14,733	136	85.83%
20	33,136	27,506	27,233	273	82.19%
Worflow 3					
1	1917	1725	1721	4	89.78%
5	8092	6998	6977	21	86.22%
10	17,165	14,303	14,264	39	83.10%
20	33,136	25,573	25,476	97	76.88%
Workflow with SimpleBlobDetector algorithm					
1	1917	1907	1905	2	99.37%
5	8092	8052	8037	15	99.32%
10	17,165	17,075	17,047	28	99.31%
20	33,136	33,004	32,954	50	99.45%

## Data Availability

The data sets used or analyzed during the current study are available from the corresponding author on reasonable request.

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
