# Peer review of "Image Segmentation and Quantification of Droplet dPCR Based on Thermal Bubble Printing Technology"

_sensors, 2022, doi:10.3390/s22197222_

Round 1

Reviewer 1 Report

Paper is interesting and novel. I would recommend to accept for publication in the present form.

Author Response

Thanks a lot for reviewer's approval.

Reviewer 2 Report

Zhu et al, applied the one of the blob detection algorithm to identify and localize droplets in bright field images and suppress bright spots and scratches when performing droplet location identification. The paper tackles an interesting and important topic. 

A few comments are: 

1. Is it possible to generate droplet directly into a reading chip, without manual transfer process.

2. I believe there are commercialized readers. Have you compared your results with these products and some other reported papers.

Overall, the authors have well organized the paper, however, I am wondering how this algorithm can improve the actual reader's sensitivity.

Reviewer 3 Report

In this paper, the authors propose an algorithm to identify and localize droplets in a ddPCR assay. The authors suppress bright spots and scratches in the bright field image and distinguished the positive droplets from negative ones accurately.

The paper is generally convincing and easy to follow. However, there are several concerns that need to be addressed,

       The authors use a ‘S’ shape channel to display droplets and it looks the width of the channel is bigger than the droplet diameter. Is there a specific reason for choosing this S shape device, as well as choosing this specific width?

       From the workflow, the authors show that the PCR reaction is performed in a reading chip in a PCR machine. Is there a problem with drop merging and evaporation? If yes, how to overcome these problems when you do image analysis?

       Some text in the manuscript needs to be improved, i.e., ul in line 21 should be l.

       The label font in the figures is too small to follow.
